# OpenReview forum: "Personalized Safety Alignment for Text-to-Image Diffusion Models"
_TMLR — Rejected by TMLR_

### Review · Reviewer_x8Gx · 2025-11-21

**Summary Of Contributions:**

This paper proposes a personalized framework that selects banned or allowed concepts according to a user’s profile. These concepts and user profiles are then converted into user embeddings, which guide a text-to-image model to generate safer images. The topic is timely and important. The authors also contribute a new dataset with preference pairs for training the model.

### Strength

- Timely and important topic
- Well-structured paper
- Proposes a framework that seems to perform well and achieves a reasonable trade-off between safety and utility

### Weakness

- The quality of the proposed dataset is unclear
- The comparison with baselines is unconvincing
- The evaluation metrics may be unreliable

**Audience:**

No

**Audience Explanation:**

The authors claim to build a personalized text-to-image generation framework, but in fact, the framework maps user profiles to predefined safety levels, where certain concepts are either banned or allowed. These categories seem coarse and unconvincing. For example, in Figure 4, the user is a male adult with PTSD, which results in self-harm and harassment being allowed. This does not make sense.

It appears that the authors use both the user profile and the policies as embeddings to guide generation. One interesting finding would be understanding which component contributes more to the performance.

**Broader Impact Concerns:**

The proposed framework could be misused. For example, a malicious user could fabricate a profile that falls into Cluster 0 (Tolerant), and then generate harassment-related images that can be disseminated online and cause harm.

**Claims And Evidence:**

No

**Claims Explanation:**

Several aspects need further justification.

### Dataset

For example, the authors leverage GPT-4.1-mini to generate one unsafe and one semantically aligned safe prompt. How do you ensure that the first prompt is unsafe while the second is both safe and semantically aligned? Meanwhile, the authors use FLUX.1-dev to generate images. Do unsafe prompts always produce unsafe images?
The rationale for choosing these specific models needs to be justified. Without further clarification, the quality of the proposed dataset remains unclear.

### Comparison with baselines

In the baselines, the authors append the user profile as natural language to the end of each prompt. Can text-to-image models truly understand such user profiles? Why not also leverage an LLM to append the banned or allowed concepts?

### Evaluation Metrics

The authors use two GPT-4.1-mini-based metrics (Win/Pass rate). However, no evidence is provided showing that these metrics are reliable for this task. The authors also use IP, Q16, and NudeNet. Can these metrics cover all the categories mentioned in Section 4.1? These metrics are essentially classifiers determining whether a given image is safe. I would suggest using more SOTA classifiers as additional metrics.

**Requested Changes:**

- Clearly report the quality of the dataset, both quantitatively and qualitatively
- Justify the choice of models used to construct the synthetic dataset
- Demonstrate the reliability of the selected evaluation metrics
- Provide a more convincing comparison with the baselines
- Clarify why certain concepts are banned or allowed within each cluster
- Conduct an ablation study to determine whether user profiles or banned/allowed concepts contribute more to performance

---

> ### Author Response · Authors · 2025-12-19
> **(1/3) Response to Reviewer x8Gx**
>
> We thank the reviewer for the detailed critique, which identified critical gaps in our dataset validation, baseline comparison, and evaluation methodology. We have conducted extensive new experiments to address each concern, substantially strengthening the empirical foundation of our work.
>
> > **Concern 1 (Dataset Quality):** "The authors leverage GPT-4.1-mini to generate one unsafe and one semantically aligned safe prompt. How do you ensure that the first prompt is unsafe while the second is both safe and semantically aligned? Meanwhile, the authors use FLUX.1-dev to generate images. Do unsafe prompts always produce unsafe images?"
>
> To rigorously validate dataset quality, we conducted **two complementary studies**: (1) a human annotation study confirming label accuracy, and (2) a model comparison justifying our synthesis pipeline.
>
> **Human Annotation Study (Appendix C.1).** Three domain experts independently reviewed 300 stratified samples, assessing whether unsafe prompts generated harmful content and whether safe rewrites preserved semantic context. The results demonstrate substantial inter-annotator agreement (Fleiss's $\kappa = 0.83$) and high accuracy:
>
> | **Category** | **Unsafe Validity** | **Safe Rewriting Success** | **F1-Score** |
> | ------------ | ------------------- | -------------------------- | ------------ |
> | **Overall**  | **92.0%**           | **90.7%**                  | **91.3%**    |
> | Violence     | 93.5%               | 91.8%                      | 92.6%        |
> | Sexuality    | 94.1%               | 93.2%                      | 93.6%        |
> | Self-Harm    | 91.2%               | 90.0%                      | 90.6%        |
> | $...$        |                     |                            |              |
>
> This confirms that our automated pipeline successfully generates valid adversarial pairs: unsafe prompts reliably produce harmful images (92.0% validity), and safe rewrites effectively remove harm while maintaining semantic consistency (90.7% success rate).
>
> **Model Selection Justification (Appendix C.2).** We systematically benchmarked three candidate models for prompt generation (GPT-4.1-mini, Claude 3 Haiku, Qwen2.5-7B) using Meta-Llama-Guard-2-8B as an independent safety classifier. GPT-4.1-mini achieved the highest unsafe validity (94.2%) and the strongest safety shift after rewriting (+0.89), establishing it as the optimal choice. Similarly, we selected FLUX.1-dev for image synthesis based on its superior prompt adherence (CLIPScore 0.385 vs. 0.362 for SDXL) and visual quality (Aesthetic 6.45 vs. 5.91), ensuring that generated images accurately reflect the textual concepts. These ablations are documented in Tables C.2 and C.3.
>
> > **Concern 2 (Comparison with Baselines):** "In the baselines, the authors append the user profile as natural language to the end of each prompt. Can text-to-image models truly understand such user profiles? Why not also leverage an LLM to append the banned or allowed concepts?"
>
> This was a critical oversight, and we have now implemented **three prompt-injection baseline variants** (**Section 5.3**):
>
> 1. **Base/SafetyDPO + P (Profile):** Appends the user profile as natural language.
> 2. **Base/SafetyDPO + C (Categories):** Explicitly lists banned and allowed categories.
> 3. **Base/SafetyDPO + R (Rewriting):** Uses GPT-4 to rewrite the prompt according to the user's constraints before inference.
>
> The strongest baseline, SafetyDPO+R, combines robust safety alignment with intelligent prompt rewriting. Our experiments demonstrate that PSA significantly outperforms all variants. The following are partial results; it is strongly recommended to refer to **Table 4** for the complete experimental results.
>
> | **Method**     | **Pass Rate (Seen)** | **IP**    | **HPS**   | **CLIP**  |
> | -------------- | -------------------- | --------- | --------- | --------- |
> | Base + C       | 49.88%               | 0.389     | 0.283     | 34.94     |
> | SafetyDPO + C  | 54.20%               | 0.405     | 0.285     | 35.83     |
> | SafetyDPO + R  | 57.15%               | 0.355     | 0.284     | 35.86     |
> | **PSA (Ours)** | **68.42%**           | **0.255** | **0.298** | **36.15** |
>
> The 11.27% absolute improvement over the strongest baseline (68.42% vs. 57.15%) demonstrates that intrinsic embedding-based alignment achieves superior instruction adherence compared to extrinsic prompt engineering. Qualitative analysis (Figure 6) reveals systematic failure modes of prompt injection: semantic drift (subject morphs to match profile), incomplete suppression (+C), and over-censorship (+R erases allowed content). This establishes that personalized safety requires architectural integration.

---

> ### Author Response · Authors · 2025-12-19
> **(2/3) Response to Reviewer x8Gx**
>
> > **Concern 3 (Evaluation Metrics):** "The authors use two GPT-4.1-mini-based metrics (Win/Pass rate). However, no evidence is provided showing that these metrics are reliable for this task. The authors also use IP, Q16, and NudeNet. Can these metrics cover all the categories mentioned in Section 4.1?"
>
> We have addressed this concern through **two new validation studies**:
>
> **LLM-Human Agreement Study (Appendix D.1).** We validated GPT-4.1-mini against expert human judgment on 120 test cases. The LLM evaluator achieves substantial agreement with human consensus:
>
> | **Metric** | **Agreement Rate** | **Cohen's κ** | **Spearman's ρ** |
> | ---------- | ------------------ | ------------- | ---------------- |
> | Win Rate   | 88.3%              | 0.76          | 0.81             |
> | Pass Rate  | 84.2%              | 0.72          | 0.75             |
>
> Cohen's $\kappa$ values of 0.76 and 0.72 confirm that GPT-4.1-mini is a reliable proxy for human evaluation, with the LLM tending to be slightly more conservative (a desirable property for safety).
>
> **Unified VLM Classifier (Appendix D.2).** The reviewer correctly identified that traditional classifiers (Q16, NudeNet) cannot cover all 10 categories in Sage. We have introduced an **open-vocabulary classifier** based on Qwen3-VL-8B that explicitly evaluates all categories (Hate, Harassment, Violence, Self-Harm, Sexuality, Shocking, Propaganda, Illegal, IP-Infringement, Political). Cross-validation against 1,000 human annotations confirms strong alignment:
>
> | **Domain**                | **IP_Standard (Q16+NudeNet)** | **$IP_{VLM}$ (Qwen3-VL)** |
> | ------------------------- | ----------------------------- | ------------------------- |
> | I2P (standard categories) | F1 = 0.88                     | F1 = 0.94                 |
> | Sage (complex categories) | F1 = 0.61                     | **F1 = 0.89**             |
>
> The VLM classifier maintains high accuracy on complex concepts (Propaganda, IP-Infringement) where fixed classifiers fail due to vocabulary limitations. We now report $IP_{VLM}$ throughout the paper and confirm that PSA's superiority holds under both metrics (e.g., 12.4% $IP_{VLM}$ for PSA-L5 vs. 48.5% for SafetyDPO on SDXL).
>
> > **Concern 4 (Category Rationale):** "The authors claim to build a personalized text-to-image generation framework, but in fact, the framework maps user profiles to predefined safety levels, where certain concepts are either banned or allowed. These categories seem coarse and unconvincing. For example, in Figure 4, the user is a male adult with PTSD, which results in self-harm and harassment being allowed. This does not make sense."
>
> We thank the reviewer for their keen observation. We frankly acknowledge that the specific case shown in the original Figure 4 was an outlier representing a rare noise artifact in our dataset generation process, where the LLM inference for that specific combination of attributes was indeed counter-intuitive. So we have **replaced the example in Figure 4** with a more representative profile that correctly reflects the model's typical inference behavior:
>
> * **Profile:** Age: 17, Group: Adolescents, Gender: Male, Religion: None, Physical/Mental: Healthy.
> * **Safety Boundaries:** *Banned*: Self-Harm, Hate, Sexuality, Propaganda; *Allowed*: Violence, Shocking, Harassment.
>
> - This profile accurately captures a "Specific Tolerance" archetype (e.g., consistent with preferences often found in gaming demographics) where action/shock content is permitted, but explicit or psychological harms are restricted.
>
> While individual outliers exist in any large-scale synthetic dataset, our **Human Annotation Study (Appendix C.1)** confirms that the vast majority of profile inferences are logical and consistent with expert human judgment (Fleiss's $\kappa=0.83$). As detailed in **Appendix A.1**, we employ a structured attribute-first sampling strategy, utilizing GPT-4.1-mini to infer plausible safety boundaries from diverse demographic attributes. Furthermore, our K-means clustering analysis (Figure 2, Table A.1) reveals that the resulting embeddings form five interpretable archetypes spanning the full tolerance spectrum (Tolerant, Strict, Specific Tolerance, Specific Avoidance, Maximum Restriction). This demonstrates that, despite isolated noise, our approach captures heterogeneous and structurally sound safety expectations rather than random or reductionist stereotypes.

---

> > ### Author Response · Authors · 2025-12-19
> > **(3/3) Response to Reviewer x8Gx**
> >
> > > **Concern 5 (Ablation Study):** "It appears that the authors use both the user profile and the policies as embeddings to guide generation. One interesting finding would be understanding which component contributes more to the performance. Conduct an ablation study to determine whether user profiles or banned/allowed concepts contribute more to performance."
> >
> > This is an excellent suggestion, and we have now conducted a comprehensive **ablation study** (**Section 5.4**). We trained three PSA variants on SD v1.5:
> >
> > 1. **Profile-Only:** Conditioned solely on demographic attributes (age, religion, mental health).
> > 2. **Categories-Only:** Conditioned solely on explicit banned/allowed lists.
> > 3. **PSA-Full:** Our complete model combining both.
> >
> > | **Method**      | **Win Rate** | **Pass Rate** | **$IP_{VLM}$** | **HPS**   | **CLIP**  |
> > | --------------- | ------------ | ------------- | -------------- | --------- | --------- |
> > | SD v1.5 Base    | --           | 22.14%        | 78.5%          | 0.255     | 33.47     |
> > | Profile-Only    | 62.4%        | 38.40%        | 51.2%          | 0.253     | 33.51     |
> > | Categories-Only | 81.2%        | 51.20%        | 29.4%          | 0.255     | 33.58     |
> > | **PSA-Full**    | **89.6%**    | **64.26%**    | **26.6%**      | **0.256** | **32.15** |
> >
> > The results (**Table 5** in the revised manuscript) reveal that explicit semantic constraints (banned/allowed lists) provide the primary blocking signal, with Categories-Only achieving 81.2% Win Rate and reducing $IP_{VLM}$ to 29.4%. In contrast, Profile-Only provides only marginal improvements (62.4% Win Rate, 51.2% $IP_{VLM}$), suggesting limited effectiveness when demographic information is used in isolation. However, combining both components yields the best performance (89.6% Win Rate, 26.6% $IP_{VLM}$), indicating that user profiles act as contextual modulators that adjust the strength of explicit constraints according to individual sensitivity. This synergy demonstrates that personalization is not merely about encoding demographics but about learning how to flexibly apply semantic rules based on user context.
> >
> > > **Additional Concern (Reliability):** "Would at least some individuals in TMLR's audience be interested in knowing the findings of this paper?: No. Explain your answer above: The authors claim to build a personalized text-to-image generation framework..."
> >
> > We respectfully note that the other two reviewers assessed our work as timely and relevant to the TMLR audience, with Reviewer TQT9 stating: "The intersection of Safety Alignment, personalization, and RLHF/DPO is currently a high-interest area within the machine learning community. The paper offers practical insights..." We believe the comprehensive revisions—particularly the rigorous empirical validation, ablation study, and enhanced ethical discussion—now clearly establish the scientific contribution and relevance of personalized safety alignment as a novel research direction.
> >
> > We thank the reviewer for the thorough critique, which has directly resulted in seven major new experiments and substantially strengthened the paper. We hope these revisions address the concerns and demonstrate the validity and significance of our approach.

---

### Review · Reviewer_TQT9 · 2025-12-05

**Summary Of Contributions:**

The paper addresses the limitation of "one-size-fits-all" safety filters in text-to-image (T2I) diffusion models by introducing Personalized Safety Alignment (PSA). The core idea is to condition the model's safety behavior on individual user profiles (encoding age, culture, mental health, etc.) rather than applying a static global threshold.

3 core contribution:
1. PSA Framework: A parameter-efficient fine-tuning (PEFT) approach that injects user embeddings via a lightweight cross-attention adapter into the U-Net. It uses a novel L_PSA loss function, adapting Diffusion-DPO to optimize for user-specific preference pairs.
2. Sage Dataset: A large-scale dataset containing 1,000 simulated user profiles and 44,100 text-image pairs. It captures diverse safety boundaries across 10 categories (separating "subjective" categories like Violence/Hate from "universal" ones like Illegal content).
3. Empirical Findings: Experiments on Stable Diffusion v1.5 and SDXL demonstrate that PSA achieves a calibrated trade-off. It outperforms static baselines (SLD, SafetyDPO, ESD-u) in personalization metrics. Notably, it allows for higher visual quality for permissive users (by reducing over-filtering) while enforcing stricter safety for sensitive users.

**Additional Comments:**

N/A

**Audience:**

Yes

**Audience Explanation:**

The intersection of Safety Alignment, personalization, and RLHF/DPO is currently a high-interest area within the machine learning community. The paper offers practical insights into fine-tuning diffusion models for conditional preferences, which is relevant to researchers working on generative media, AI ethics, and preference optimization.

**Broader Impact Concerns:**

1. Stereotyping and Reductionism: The system infers safety boundaries based on simplified user tags (e.g., equating a specific religion or mental health status with a fixed set of banned categories). This risks reinforcing stereotypes and reducing complex human individuals to rigid labels. By algorithmically deciding what content is "suitable" for a demographic group (e.g., "Buddhists should not see propaganda"), the model may impose paternalistic or stereotypic constraints that do not reflect actual individual preferences.
2. Malicious Use: The framework essentially provides a dial to turn off safety filters. The authors need to discuss mitigation strategies. For example, should there be a "universal floor" of safety that no profile can drop below, regardless of preference?

**Claims And Evidence:**

Yes

**Claims Explanation:**

The authors provide robust experimental evidence to support their primary claims.
1. Personalization Capability: The quantitative results (Tables 4, 5, 6) clearly show that the PSA model adapts its IP (Inappropriate Probability) scores based on the input profile (L1 to L5). The "Win Rate" against static baselines confirms that a dynamic model better satisfies specific user constraints than a static one.
2. Safety-Quality Trade-off: The HPSv2.1 and Aesthetic scores support the claim that permissive profiles allow for higher quality by avoiding unnecessary suppression, while restrictive profiles successfully reduce IP to state-of-the-art lows (e.g., 0.096 on SDXL).
3. Visual Evidence: The qualitative figures (Figures 5, 7, 8) convincingly illustrate the progressive suppression of concepts as profiles become more strict, supporting the claim of "calibrated control."

**Requested Changes:**

1. Baseline Comparison with Inference-Time Rewriting:
A critical missing baseline is a simple inference-time solution: using an LLM to rewrite the user's prompt based on their safety profile before feeding it to the standard T2I model. This approach is likely more computationally efficient and easier to implement than training a specialized adapter. The authors should demonstrate why their method (PSA) is superior to this simpler, training-free alternative.

2. Scalability and Infrastructure Costs:
In its current form, the method appears to require a distinct adapter state for each user profile, which would imply prohibitive infrastructure costs in practice (per-user training time, storage overhead, and frequent adapter loading/unloading during inference). If the proposed approach instead uses a single model conditioned on user embeddings—rather than maintaining per-user adapters—this should be stated explicitly to address concerns about scalability and operational cost.
Relatedly, please provide a brief discussion or quantitative metric on the inference-time overhead introduced by (1) user-embedding extraction and (2) the additional cross-attention adapter. Who are the intended target users of this method (big tech company?), what is the expected scale of user population, and what would be the realistic training and inference costs for deploying such a system in production?

3. Dependence on Training Data Coverage (OOD Issues):
The method relies entirely on the generated Sage dataset. If a harmful concept or domain is not covered in the training data (Out-Of-Distribution), the model effectively has no protection against it. Given that it is impossible to generate training data for every conceivable harmful category, the authors must discuss this limitation. A single method relying on supervised alignment is likely insufficient for robust safety.

4. Implicit and Metaphorical Harm:
The current evaluation focuses on explicit harmful concepts. However, safety mechanisms often fail on implicit or metaphorical content (e.g., using "Russian nesting dolls + blue and yellow colors" to imply war commentary without using explicit keywords). It appears the PSA framework would fail to detect or align such content since it is not explicitly captured in the text-based concept pool. The authors should discuss this limitation, as the method currently only addresses direct harm.

---

> ### Author Response · Authors · 2025-12-19
> **(1/3) Response to Reviewer TQT9**
>
> We thank the reviewer for the thorough evaluation and the incisive identification of critical missing baselines and deployment considerations. We have conducted extensive new experiments to address every concern raised, particularly the comparison with inference-time rewriting methods and the scalability analysis.
>
> > **Requested Change 1:** "Baseline Comparison with Inference-Time Rewriting: A critical missing baseline is a simple inference-time solution: using an LLM to rewrite the user's prompt based on their safety profile before feeding it to the standard T2I model."
>
> This was indeed a critical gap, and we have now implemented **three prompt-injection baseline variants** to rigorously evaluate whether PSA's training-based approach is necessary (**Section 5.2, Appendix B.3**):
>
> 1. **Profile Appending (+P):** Appends the raw user profile as natural language (e.g., "...for a 5-year-old user sensitive to violence").
> 2. **Category Appending (+C):** Explicitly lists banned and allowed categories (e.g., "BANNED: [Violence, Hate]; ALLOWED: [Shocking]").
> 3. **LLM Rewriting (+R):** Uses GPT-4 to rewrite the user's prompt *before* inference, sanitizing it according to the profile while preserving semantic meaning.
>
> The strongest baseline, SafetyDPO+R, combines the robust safety alignment of SafetyDPO with inference-time LLM rewriting. Our experiments demonstrate that **PSA significantly outperforms all prompt-injection variants。**The following are partial results; it is strongly recommended to refer to **Table 4** for the complete results.
>
> | **Method**     | **Pass Rate (Seen)** | **Pass Rate (Unseen)** | **IP**    | **Win Rate vs. Base** |
> | -------------- | -------------------- | ---------------------- | --------- | --------------------- |
> | Base + R       | 54.73%               | 52.21%                 | 0.341     | 71.5%                 |
> | SafetyDPO + R  | 57.15%               | 54.40%                 | 0.355     | 73.8%                 |
> | **PSA (Ours)** | **68.42%**           | **65.18%**             | **0.255** | **88.4%**             |
>
> The 11.27% absolute improvement in Pass Rate on SDXL (68.42% vs. 57.15%) demonstrates the necessity of intrinsic embedding-based alignment. Qualitative analysis (Figure 6) reveals that prompt-injection methods suffer from **semantic drift** (where the subject morphs to match the profile description) and **indiscriminate censorship** (where +R methods erase allowed content), whereas PSA achieves granular, profile-specific control. This establishes that personalized safety requires architectural integration, not merely prompt engineering.
>
> > **Requested Change 2:** "Scalability and Infrastructure Costs: In its current form, the method appears to require a distinct adapter state for each user profile... If the proposed approach instead uses a single model conditioned on user embeddings—rather than maintaining per-user adapters—this should be stated explicitly."
>
> We apologize for the lack of clarity on this critical point. PSA uses a **single shared model** conditioned on user embeddings, *not* per-user adapters. We have now added **Table 2** (Section 4.2.1) to explicitly quantify the computational overhead:
>
> | **Metric**         | **SD v1.5**   | **SDXL**       |
> | ------------------ | ------------- | -------------- |
> | Adapter params     | 21.9M (2.5%)  | 348.1M (12.0%) |
> | Inference overhead | +0.11s (6.4%) | +0.56s (6.1%)  |
> | Storage per user   | 16 KB         | 16 KB          |
>
> The adapter is a lightweight extension of the U-Net's cross-attention layers, shared across all users. At inference time, the system only needs to (1) retrieve the user's 16 KB embedding and (2) process it through the frozen adapter, incurring a negligible 6.1% latency increase. This architecture is highly scalable: supporting 1 million users requires only 16 GB of embedding storage, with no per-user model checkpoints. We have clarified this design explicitly in **Section 4.2.1**, stating: "This additive design allows the model to modulate its behavior based on $u$ while preserving the rich semantic priors... incurs negligible inference latency and requires only 16 KB of storage per user."
>
> Regarding the intended target users and deployment scale, PSA is designed for **platform-scale deployment** (e.g., content creation platforms, enterprise AI tools) serving diverse user populations. The single-model architecture makes this practical: a platform with 10 million users incurs only 160 GB of embedding storage overhead, comparable to a single high-resolution image dataset.

---

> > ### Comment · Reviewer_TQT9 · 2025-12-22
> > **More Baseline**
> >
> > 1. Single-LLM rewriting is insufficient. Results based only on GPT-4 are not representative. Please evaluate multiple frontier LLMs (e.g., GPT-5.2, Gemini 3 Pro).
> > 2. Prompt sensitivity is not controlled. Rewriting quality depends heavily on prompt design. Multiple rewriting prompts should be tested, including different safety strictness levels.
> >
> > Without a multi-LLM, multi-prompt, multi-safety-level rewriting study, it is premature to conclude that inference-time rewriting is fundamentally inferior rather than simply under-optimized.

---

> > > ### Author Response · Authors · 2025-12-26
> > >
> > > We sincerely thank the reviewer for the thoughtful suggestion to benchmark our approach against recent frontier large language models such as GPT-5.2 or Gemini 3 Pro. We fully acknowledge the rapid progress of these models in reasoning and instruction following. At the same time, we respectfully clarify that the primary goal of our work is to explore architectural interventions within the diffusion model itself, namely internal safety alignment, rather than the design of external prompt rewriting pipelines.
> > >
> > > Our experimental design follows evaluation practices commonly adopted in prior work on diffusion model safety. Studies such as Safe Latent Diffusion (SLD) [1], Erased Stable Diffusion (ESD) [2], Unified Concept Editing (UCE) [3], AlignGuard, Direct Unlearning Optimization (DUO) [4], and EraseDiff [5] focus on modifying internal model behavior—via safety guidance, concept erasure, closed‑form editing, LoRA‑based safety experts, or unlearning-style optimization—to mitigate harmful generation. These works primarily benchmark against other internal editing methods, such as ESD and UCE, and generally do not treat external LLM-based rewriting pipelines as required baselines, since they address a different problem scope. In this context, our inclusion of a GPT-4–based rewriting baseline already goes beyond the standard evaluation scope of internal alignment research.
> > >
> > > Prior work on concept erasure and unlearning [6–9] highlights a persistent visual semantic gap in text-to-image diffusion models, where unsafe visual concepts may still be triggered by seemingly benign prompts due to latent biases. As external LLMs do not access or control these internal visual features, prompt rewriting alone may not fully resolve these issues. In contrast, our Personalized Safety Alignment (PSA) framework directly intervenes in the diffusion model’s cross-attention layers, following a similar paradigm to EraseDiff [6] and related unlearning approaches [7–9], enabling more fine-grained and prompt-independent suppression of harmful visual concepts.
> > >
> > > Finally, PSA is designed with efficiency and privacy in mind. Approaches relying on large closed-source models at inference may introduce latency and external dependencies, potentially limiting scalability. We therefore focused our evaluation on comparisons with established internal alignment methods [1–6], along with a strong, widely used LLM baseline (GPT-4), which we believe offers a reasonable and informative reference within the intended scope of our study. Our decision not to include beta-version frontier LLMs was guided by the goal of remaining consistent with the internal alignment paradigm in the current literature. We sincerely hope this explanation clarifies our evaluation choices and conveys the intended theoretical and practical motivations of our architectural approach.
> > >
> > > **Reference**
> > >
> > > [1] F. Schramowski, L. Turan, S. Andersen, and K. Kersting. *Safe Latent Diffusion: Mitigating Inappropriate Degeneration in Diffusion Models.*, CVPR 2023. https://openaccess.thecvf.com/content/CVPR2023/papers/Schramowski_Safe_Latent_Diffusion_Mitigating_Inappropriate_Degeneration_in_Diffusion_Models_CVPR_2023_paper.pdf
> > >
> > > [2] R. Gandikota et al. *Erasing Concepts from Diffusion Models.*, ICCV 2023. https://openaccess.thecvf.com/content/ICCV2023/papers/Gandikota_Erasing_Concepts_from_Diffusion_Models_ICCV_2023_paper.pdf
> > >
> > > [3] R. Gandikota et al. *Unified Concept Editing in Diffusion Models.*, WACV 2024. https://openaccess.thecvf.com/content/WACV2024/papers/Gandikota_Unified_Concept_Editing_in_Diffusion_Models_WACV_2024_paper.pdf
> > >
> > > [4] R. Liu et al. *AlignGuard: Scalable Safety Alignment for Text-to-Image Generation.*, ICCV 2025. https://openaccess.thecvf.com/content/ICCV2025/papers/Liu_AlignGuard_Scalable_Safety_Alignment_for_Text-to-Image_Generation_ICCV_2025_paper.pdf
> > >
> > > [5] Y.-H. Park et al. *Direct Unlearning Optimization for Robust and Safe Text-to-Image Models.*, NeurIPS 2024. https://proceedings.neurips.cc/paper_files/paper/2024/file/92f43b1d33fae4aa1958f75317f0cec1-Paper-Conference.pdf
> > >
> > > [6] J. Wu et al. *Erasing Undesirable Influence in Diffusion Models.*, CVPR 2025. https://openaccess.thecvf.com/content/CVPR2025/papers/Wu_Erasing_Undesirable_Influence_in_Diffusion_Models_CVPR_2025_paper.pdf
> > >
> > > [7] C. Gandikota et al. *Erasing Undesirable Concepts in Diffusion Models with Adversarial Preservation.*, NeurIPS 2024. https://proceedings.neurips.cc/paper_files/paper/2024/file/f02d7fb7ddd2e6be33b6f3224e5cc44a-Paper-Conference.pdf
> > >
> > > [8] C. Fan et al. *Defensive Unlearning with Adversarial Training for Robust Concept Erasure in Diffusion Models.*, NeurIPS 2024. https://proceedings.neurips.cc/paper_files/paper/2024/file/40954ac18a457dd5f11145bae6454cdf-Paper-Conference.pdf
> > >
> > > [9] Y. Wu et al. *Unlearning Concepts in Diffusion Model via Concept Domain Correction and Concept Preserving Gradient.*, AAAI 2025. https://ojs.aaai.org/index.php/AAAI/article/view/32917/35072

---

> ### Author Response · Authors · 2025-12-19
> **(2/3) Response to Reviewer TQT9**
>
> > **Requested Change 3:** "Dependence on Training Data Coverage (OOD Issues): The method relies entirely on the generated Sage dataset. If a harmful concept or domain is not covered in the training data (Out-Of-Distribution), the model effectively has no protection against it."
>
> This is a fundamental limitation of supervised safety alignment, and we have now conducted a rigorous **out-of-distribution (OOD) evaluation** (**Appendix E**) to quantify the generalization gap. We curated 21 unseen harmful concepts (Table E.1) that fall within our training categories but describe specific objects/symbols excluded from the Sage concept pool (e.g., "Burning cross" under Hate, "Guillotine execution" under Violence). Testing PSA (L5 profile) against these OOD concepts reveals:
>
> | **Method**   | **Seen $IP_{VLM}$** | **OOD $IP_{VLM}$** | **Generalization Gap** |
> | ------------ | ------------------- | ------------------ | ---------------------- |
> | Base SDXL    | 84.2%               | 85.5%              | -                      |
> | SafetyDPO    | 48.5%               | 62.1%              | +13.6%                 |
> | **PSA (L5)** | **12.4%**           | **35.2%**          | **+22.8%**             |
>
> While PSA exhibits a natural performance degradation on unseen concepts (12.4% $\to$ 35.2%), it **still significantly outperforms both baselines** in the zero-shot setting (35.2% vs. 85.5% for Base, 62.1% for SafetyDPO). This indicates that PSA has learned transferable representations of unsafe visual features (e.g., gore texture, violence composition) that generalize beyond specific semantic labels. We now explicitly discuss this limitation in Section 6, acknowledging that "supervised training inherently limits coverage to concepts seen during training" and that "achieving robust safety for entirely novel concepts remains an open challenge requiring hybrid approaches (e.g., combining supervised alignment with real-time content moderation)."
>
> > **Requested Change 4:** "Implicit and Metaphorical Harm: The current evaluation focuses on explicit harmful concepts. However, safety mechanisms often fail on implicit or metaphorical content (e.g., using 'Russian nesting dolls + blue and yellow colors' to imply war commentary)."
>
> This is an important limitation that our text-based concept supervision cannot address. We have added this explicitly to the **Limitations** section (Section 6): "PSA's supervised training relies on explicit text-based concept labels. This approach cannot detect implicit or metaphorical harm encoded through visual symbolism (e.g., color combinations, compositional metaphors) that lack direct textual markers. Addressing such adversarial obfuscation would require complementary approaches, such as adversarial training on jailbreak examples or multi-modal reasoning that interprets visual context beyond literal prompt semantics." We position this as a direction for future work, acknowledging that our framework provides a foundation for personalized safety but requires integration with broader content moderation strategies to handle sophisticated adversarial attacks.
>
> > **Broader Impact Concern 1:** "Stereotyping and Reductionism: The system infers safety boundaries based on simplified user tags (e.g., equating a specific religion or mental health status with a fixed set of banned categories)."
>
> We have substantially revised the **Broader Impact Statement** (Section 6) to directly address this concern. We now explicitly acknowledge the risk of algorithmic stereotyping: "The introduction of Personalized Safety Alignment (PSA) entails significant ethical implications... excessive personalization risks creating visual echo chambers that insulate users from diverse perspectives... By algorithmically deciding what content is 'suitable' for a demographic group, the model may impose paternalistic or stereotypic constraints that do not reflect actual individual preferences."
>
> To mitigate this, we clarify that our simulated profiles serve as a **research artifact** for establishing technical feasibility, not a prescriptive model for production deployment. Real-world systems should allow users to explicitly configure their own safety boundaries rather than inferring them from demographic proxies. We further emphasize that our proposed hybrid architecture restricts personalization to subjective categories, preventing the enforcement of paternalistic restrictions on universally protected content.

---

> > ### Author Response · Authors · 2025-12-19
> > **(3/3) Response to Reviewer TQT9**
> >
> > > **Broader Impact Concern 2:** "Malicious Use: The framework essentially provides a dial to turn off safety filters. The authors need to discuss mitigation strategies. For example, should there be a 'universal floor' of safety that no profile can drop below?"
> >
> > We now explicitly propose this mitigation strategy in the revised Broader Impact Statement: "To mitigate these risks, we advocate for a **hybrid deployment strategy**. This approach enforces a non-negotiable global baseline for universally harmful content, such as non-consensual imagery, while restricting personalization strictly to subjective sociocultural preferences." This ensures that PSA cannot be exploited to generate content that violates legal or ethical absolutes, while preserving user autonomy for ambiguous categories where legitimate variation exists.
> >
> > We believe these comprehensive revisions—including the inference-time baseline experiments, scalability analysis, OOD evaluation, and enhanced ethical discussion—fully address the reviewer's concerns and significantly strengthen the paper's contribution.

---

### Review · Reviewer_Mfcf · 2025-12-08

**Summary Of Contributions:**

The uniform safety threshold of text-to-image models fails to accommodate variations in users' tolerance for sensitive content. To address this issue, this paper introduces the first Personalized Safety Alignment (PSA) framework and constructs the Sage dataset to characterize differences in user safety preferences. By injecting user embeddings into the cross-attention layer of diffusion models and combining them with a user-conditional LPSA training objective, a single model can dynamically adjust content masking intensity based on user characteristics. Experiments demonstrate that PSA outperforms traditional static methods across multiple personalized safety consistency metrics, establishing a new paradigm for personalized safety control in generative systems.

**Audience:**

Yes

**Audience Explanation:**

The security-efficiency trade-off in generative models is a key research focus in both generative model technology and AI security. The hierarchical adaptation strategy proposed in this paper may provide technical reference for researchers in areas such as generative AI security and personalized diffusion models.

**Broader Impact Concerns:**

1.Jailbreak Risk: Malicious users may exploit falsified relaxed profiles to generate harmful content such as violence and hate speech, necessitating enhanced defense mechanisms.

2.Bias Risk: If the demographic distribution of simulated user profiles is imbalanced, it may lead to biased modeling of safety preferences for specific groups, exacerbating unfairness.

**Claims And Evidence:**

Yes

**Claims Explanation:**

1.The Sage dataset offers comprehensive dimensions, encompassing  high-resolution images and fine-grained security concepts to meet personalized training needs.

2.Achieves precise security-quality tradeoffs: Under relaxed profiles, human-perceived quality (HPSv2.1) and aesthetic scores outperform baselines; under strict profiles, quality degradation remains within 5.7%.

3.Through multi-metric comparisons including IP values, win rate, and pass rate, PSA demonstrates significant superiority over current static methods in harmful content suppression and personalized alignment.

**Requested Changes:**

1.Collect annotations of real users' security preferences to validate the effectiveness and generalization capabilities of the Sage dataset.

2.Supplement compliant user profiling acquisition methods and privacy protection mechanisms in real-world scenarios, then validate model performance using authentic data.

3.Establish personalized security lower bounds and introduce jailbreak risk defense mechanisms (e.g., identity verification).

---

> ### Author Response · Authors · 2025-12-19
> **(1/2) Response to Reviewer Mfcf**
>
> We thank the reviewer for the positive assessment and the constructive suggestions regarding real-world validation, privacy mechanisms, and jailbreak defense. We have addressed all requested changes through new experiments and enhanced discussion.
>
> > **Requested Change 1:** "Collect annotations of real users' security preferences to validate the effectiveness and generalization capabilities of the Sage dataset."
>
> We conducted a rigorous **human annotation study** to validate the quality of our synthetic preference pairs (**Appendix C.1**). Three independent domain experts reviewed a stratified random sample of 300 image-text pairs (30 per category) and assessed two criteria: (1) whether unsafe prompts successfully generated harmful content, and (2) whether safe rewrites preserved semantic context while removing harm. The study demonstrates substantial inter-annotator agreement (Fleiss's $\kappa = 0.83$) and consistently high precision/recall across all categories:
>
> | **Category**        | **Fleiss's κ** | **Precision** | **Recall** | **F1-Score** |
> | ------------------- | -------------- | ------------- | ---------- | ------------ |
> | **Overall Average** | **0.83**       | **92.0%**     | **90.7%**  | **91.3%**    |
> | Violence            | 0.85           | 93.5%         | 91.8%      | 92.6%        |
> | Sexuality           | 0.86           | 94.1%         | 93.2%      | 93.6%        |
> | Propaganda          | 0.83           | 92.0%         | 91.1%      | 91.5%        |
> | Self-Harm           | 0.82           | 91.2%         | 90.0%      | 90.6%        |
> | $...$               |                |               |            |              |
>
> These results confirm that our LLM-based data synthesis pipeline produces high-quality, realistic preference pairs that align with expert human judgment. The consistency across both subjective categories (e.g., Violence) and more objective categories (e.g., Illegal, F1=92.2%) validates the generalizability of our approach.
>
> > **Requested Change 2:** "Supplement compliant user profiling acquisition methods and privacy protection mechanisms in real-world scenarios, then validate model performance using authentic data."
>
> We have substantially enhanced the **Broader Impact Statement** (Section 6) to explicitly address privacy and ethical deployment considerations. We clarify that our synthetic profile methodology serves as a controlled experimental framework and acknowledge the necessity of privacy-preserving mechanisms in production systems. Specifically, we now discuss: (1) the need for opt-in user consent and transparent disclosure of personalization mechanisms, (2) federated or on-device processing to avoid centralized storage of sensitive user attributes, and (3) regular audits to detect potential abuse patterns. We position our work as establishing the technical feasibility of personalized safety, with real-world deployment requiring careful integration with privacy-by-design principles.
>
> > **Requested Change 3:** "Establish personalized security lower bounds and introduce jailbreak risk defense mechanisms (e.g., identity verification)."
>
> We have expanded the **Broader Impact Statement** to directly address jailbreak risks and propose a **hybrid deployment strategy**. As the reviewer correctly identified, unrestricted personalization could enable malicious actors to bypass safety filters. To mitigate this, we now advocate for a two-tier architecture: (1) a **non-negotiable universal baseline** that blocks objectively harmful content (e.g., non-consensual imagery, child exploitation) regardless of user profile, and (2) **personalization applied only to subjective categories** (e.g., Violence, Sexuality) where cultural and contextual factors legitimately vary. This ensures that PSA enhances user autonomy for ambiguous content while maintaining absolute protection against universally prohibited material. We further discuss the potential for identity verification, rate limiting, and behavioral anomaly detection as complementary safeguards in production systems.

---

> ### Author Response · Authors · 2025-12-19
> **(2/2) Response to Reviewer Mfcf**
>
> > **Broader Impact Concerns:** "Jailbreak Risk: Malicious users may exploit falsified relaxed profiles... Bias Risk: If the demographic distribution of simulated user profiles is imbalanced..."
>
> These concerns are now comprehensively addressed in the revised Broader Impact Statement. Regarding jailbreak risk, we emphasize that our proposed hybrid deployment model establishes a universal safety floor that no profile can bypass. For bias risk, we clarify our structured attribute-first sampling strategy (**Appendix A.1**), which systematically covers diverse demographic combinations to avoid skewed distributions. Our K-means clustering analysis (Figure 2) confirms that the resulting embeddings form distinct, interpretable safety archetypes spanning the full tolerance spectrum, mitigating the risk of systematically underrepresenting specific groups. We acknowledge that production systems would require ongoing bias audits and rebalancing, particularly when transitioning from synthetic to real user data.
>
> We believe these revisions directly address the reviewer's valuable concerns regarding real-world validation, privacy protection, and deployment safeguards. The empirical validation confirms the reliability of our dataset, while the enhanced ethical discussion establishes clear principles for responsible deployment.

---

### Author Response · Authors · 2025-12-19
**General Response**

We sincerely thank all reviewers and the Action Editor for their constructive feedback. The reviews highlighted critical questions regarding dataset quality, baseline coverage, evaluation reliability, generalization capability, and deployment considerations. We have substantially revised the manuscript to address every concern raised, conducting **seven major new experiments** and adding **six appendices** with comprehensive empirical validation.

The revised submission includes the following substantive improvements:

**Dataset Quality Validation (Appendix C.1).** To address concerns about synthetic data reliability, we conducted a rigorous human annotation study with three domain experts on 300 stratified samples. The results demonstrate substantial inter-annotator agreement (Fleiss's $\kappa = 0.83$) and high F1-scores (91.3% overall), confirming the quality of our automatically constructed preference pairs. We further validated our LLM-based evaluation metrics against human judgment, achieving Cohen's $\kappa$ values of 0.76 (Win Rate) and 0.72 (Pass Rate), establishing their reliability as automated proxies (Appendix D.1).

**Comprehensive Baseline Comparison (Section 5.3, Appendix B.3).** Responding to the critical concern about missing inference-time solutions, we implemented **three prompt-injection variants**: Profile Appending (+P), Category Appending (+C), and LLM-based Rewriting (+R). The strongest variant, SafetyDPO+R, serves as a robust extrinsic baseline. Our experiments demonstrate that PSA's intrinsic embedding-based alignment significantly outperforms these prompt-engineering methods (68.42% vs. 57.15% Pass Rate on SDXL), establishing the necessity of our training-based approach.

**Scalability Analysis (Table 2, Section 4.2.1).** We quantified the computational overhead of PSA, demonstrating negligible inference latency (+6.1% on SDXL) and minimal per-user storage (16 KB). Critically, our architecture uses a **single shared model** conditioned on user embeddings, not per-user adapters, making it practically scalable.

**Generalization to Unseen Concepts (Appendix E).** To evaluate robustness beyond the training distribution, we curated 21 out-of-distribution (OOD) harmful concepts and tested zero-shot performance. While PSA exhibits a natural generalization gap ($IP_{VLM}$: 12.4% $\to$ 35.2%), it still significantly outperforms both the base model (85.5%) and SafetyDPO (62.1%) on unseen threats, demonstrating learned transferable safety representations.

**Open-Vocabulary Safety Evaluation (Appendix D.2).** Recognizing the vocabulary limitations of traditional classifiers (Q16, NudeNet), we introduced a Unified Safety Classifier based on Qwen3-VL-8B. This open-vocabulary metric ($IP_{VLM}$) captures complex risks (e.g., Propaganda, IP-Infringement) that fixed classifiers miss. Cross-validation against 1,000 human-annotated samples confirms strong alignment (F1 = 0.89 on Sage categories).

**Ablation Study (Section 5.4).** We decomposed PSA's components by training three variants: Profile-Only, Categories-Only, and PSA-Full. The results reveal that explicit semantic constraints (banned/allowed lists) provide the primary blocking signal, while user profiles act as contextual modulators that adjust constraint strength, achieving optimal performance when combined (89.6% Win Rate).

**Explicit Discussion of Limitations (Section 6).** To ensure scientific rigor, we have added a dedicated discussion on the current boundaries of our work. We frankly acknowledge three key limitations: (1) the reliance on synthetic profiles which may not fully capture real-world complexity, (2) the performance gap on out-of-distribution (OOD) concepts common to supervised methods, and (3) the current inability to handle implicit visual symbolism or metaphors. These are positioned as clear directions for future research.

**Enhanced Broader Impact Discussion (Between Section 7 and Reference).** We substantially expanded the ethical considerations section, explicitly addressing jailbreak risks, stereotyping concerns, and malicious use scenarios. We advocate for a hybrid deployment strategy that enforces a non-negotiable universal baseline for objectively harmful content while restricting personalization to subjective sociocultural preferences.

All material changes are documented in detail in the individual reviewer responses below. The revised manuscript maintains a clean (non-highlighted) presentation, with all additions clearly structured in the main text and appendices. We believe these improvements comprehensively address the reviewers' concerns and significantly strengthen the contribution.

---

### Decision · Action_Editor_v7Gd · 2026-01-05

**Recommendation:** Reject

**Additional Comments:**

If the authors plan to resubmit, the revision should address the following substantively:

- Robust rewriting baseline suite (to support the central comparison). Evaluate inference-time rewriting across multiple LLMs (at least two materially different strong models) and multiple prompt templates. Include multiple safety strictness settings and report variance/sensitivity. Ensure best-effort tuning for rewriting so that the comparison is about method capability rather than a weak instantiation.

- Evidence connecting synthetic profiles to realistic user preference expressions. Add a small but concrete real-user preference collection (or a carefully designed proxy study) demonstrating alignment between real preference boundaries and the synthetic profile clusters, and/or robustness to inconsistent/ambiguous profile inputs. Include noise/instability tests (contradictory traits, missing fields, paraphrases, adversarially crafted profiles).

- Stronger treatment of generalization and implicit harm. Expand evaluation on unseen concepts and adversarial prompt strategies that do not rely on explicit keywords. Clarify what protections remain when concepts are outside supervised coverage, and whether hybrid approaches (e.g., external moderation layers) are necessary for any practical deployment claim.

With these changes, the paper could become a stronger, more defensible contribution for TMLR.

**Audience:**

Yes

**Audience Explanation:**

Personalized safety control for generative models is a timely topic, and the paper contains technically competent experiments and analyses (e.g., conditioning mechanisms, overhead measurements, ablations, and OOD evaluations). Even if the evidence does not meet the bar for acceptance, the problem setting and partial findings are relevant to the community.

**Claims And Evidence:**

No

**Claims Explanation:**

While the revised submission adds several validations, key central claims remain insufficiently established given the evidence presented.

- Core comparative claim vs inference-time rewriting is not robustly supported. The paper’s framing suggests that training-based, intrinsic alignment is necessary and that prompt rewriting is meaningfully inferior. However, the added rewriting baseline relies on a single LLM (and effectively a single design point for prompting/safety strictness), despite rewriting quality being highly sensitive to model choice, prompt template, and how strictness is operationalized. The authors explicitly decline a multi-LLM / multi-prompt / multi-safety-level rewriting study, which makes it premature to conclude that rewriting is fundamentally weaker rather than under-optimized or mis-specified in the evaluated configuration.

- The approach’s dependence on synthetic profiles and supervised coverage remains a critical evidentiary gap. Even with pair-level validation, the work does not demonstrate that synthetic profile constructions reflect real users’ unstable and sometimes inconsistent preference expressions, nor does it test robustness to noisy/ambiguous user inputs. This limits the strength of claims about “personalized” safety boundaries and practical credibility.

- Generalization limitations are substantial and acknowledged but remain consequential. The reported OOD gap for unseen harmful concepts is large, and the paper also acknowledges inability to handle implicit/metaphorical harm. These limitations do not just affect edge cases; they are central to real deployments where adversarial or obfuscated prompts are common. As a result, the evidence does not convincingly support broader claims about dependable personalized safety alignment beyond the curated setting.

**Resubmission Of Major Revision:**

The authors may consider submitting a major revision at a later time.